# Analysis of the SARS-CoV-2 epidemic in Italy: The role of local and interventional factors in the control of the epidemic

**Daniele Lilleri**[1,2]*, **Federica Zavaglio**[1,2], **Elisa Gabanti**[1,2], **Giuseppe Gerna**[1], **Eloisa Arbustini**[1]

**1** Laboratorio Genetica–Trapiantologia e Malattie Cardiovascolari, Fondazione IRCCS Policlinico San Matteo, Pavia, Italy, **2** Microbiologia e Virologia, Fondazione IRCCS Policlinico San Matteo, Pavia, Italy

* d.lilleri@smatteo.pv.it

**Data Availability Statement:** All relevant data are within the manuscript.

**Funding:** This work was funded by Fondazione Cariplo, Milano, Italy (Grant CoVIM to DL). The

## Abstract

Containment measures have been applied in several countries in order to limit the diffusion of the SARS-CoV-2 epidemic. The scope of this study is to analyze the evolution of the first wave of the SARS-CoV-2 epidemic throughout Italy and factors associated to the different way it spread in the Italian Regions, starting from the day that the first indigenous cases were detected through day 81 (6 days after the end of the strict lockdown). Data were obtained from daily reports and are represented as number (and percentage) of cases/ 100,000 persons. A lockdown with movement restrictions, especially across Regions, was declared at day 20. At day 81, 219,070 cases (363/100,000 persons) were diagnosed. A regression analysis based on the Gompertz model predicts a total number 233,606 cases (386/100,000 persons) at the end of the epidemic. The 21 areas, divided into Italian Regions and autonomous Provinces, showed a wide range in the frequency of cases at day 81 (58– 921, median 258/100,000 persons) and total predicted cases (58–946, median 267/100,000 persons). Similarly, the predicted time for the end of the wave of the epidemic (considering as surrogate marker the time at which 99% of the total cases are predicted to occur) was highly variable, ranging from 64 to 136 (median 99) days. We analyzed the impact of local and interventional variables on the epidemic curve in each Region. The number of cases correlated inversely with the distance from the area in which first cases were detected and directly also with the gross domestic product *pro capite* (as a marker of industrial activity) of the Region. Moreover, an earlier start of the lockdown (i.e. in the presence of a lower number of cases) and wider testing were associated with a lower final number of total cases. In conclusion, this analysis shows that population-wide testing and early lockdown enforcement appear effective in limiting the spreading of the SARS-CoV-2 epidemic.

## Introduction

After the emergence in China of the novel coronavirus SARS-CoV-2 at the end of 2019 [1, 2], the coronavirus disease 2019 (COVID-19) spread worldwide as a pandemic disease. As at October 8, 36 million cases throughout the world (0.46% of the overall population) and 1

funder had no role in study design, data collection and analysis, decision to publish, or preparation of the manuscript. There was no additional external funding received for this study.

**Competing interests:** The authors have declared that no competing interests exist.

million related deaths have been reported to the World Health Organization (WHO). The epidemic showed an uneven distribution among different countries: in USA and Brazil 7.4 and 5 million cases, respectively, were reported to the WHO (corresponding to >2% of the population), in Europe 6.6 million cases (1.46% of the population), 1.2 million in Africa (0.10%), 27,000 in Australia (0.11%), and 90,000 in China (0.01%).

Although the infection is asymptomatic or mildly symptomatic in the majority of cases (>80%), a smaller proportion of patients, particularly the elderly and those with chronic diseases, develop a severe or critical disease, involving primarily the lungs but also vascular pathology and multi-organ failure.

On March 7, the WHO issued guidelines to slow down the transmission, decrease the number of cases and prevent outbreaks in the community [3]. Among the preventive measures, hand hygiene, respiratory etiquette, confinement of cases and contacts and social distancing were suggested.

Different containment measures were implemented in each country to reduce the burden of the epidemic. The Oxford COVID-19 Government Response Tracker compares the policy responses of governments around the world and throughout time [4]. The Stringency score (ranging from 0 to 100) provides a quantitative index of the level of the containment policy, taking into account school and workplace closing, restrictions on gatherings and public events, home confinement, closure of public transport, restrictions on internal and international movement, information campaigns, and whether they are applied to the general territory or to targeted regions. For example, as reported on the WHO website [5], Sweden and Japan adopted poor stringent measures (maximum stringency index reported during the epidemic: 46 and 47), USA higher but not completely stringent measures (maximum stringency index: 73), other European countries such as Spain and France applied more stringent measures (maximum index: 85, 88).

Italy was the first of the Western countries to be severely affected [6] and currently accounts for 333,940 reported cases (0.55% of the population). The first indigenous cases were diagnosed on February 20 (day 1) in Northern Italy (Lombardia Region) and subsequently the infection spread throughout the country. Infected persons that were not hospitalized were quarantined at home, as well as the close contacts of infected subjects. After initial local measures to contain the epidemic spread, a nationwide lockdown was implemented since March 10 (day 20), and subsequently, restriction reinforcements were adopted, reaching a high maximum level of the stringency index (94/100). The movement of people, especially across regional borders, was blocked (except for health emergencies or unavoidable work needs); any form of gathering and social or sport events were prohibited; all commercial and non-essential activities were closed [7]. Total lockdown continued until May 4 (day 75), when a partial re-opening was declared by the government.

The scope of the present work is to analyze as per May 10 (day 81) the evolution of the SARS-CoV-2 epidemic throughout Italy and the Italian Regions. Since the spread of the infection at the beginning of the lockdown was uneven, borders between Regions were closed, and since each Region has an autonomous health system, we could approximately consider each Region as an independent unit and analyze the potential association between local variables and the epidemic parameters. Useful information for health policymakers can arise from this analysis, which shows the importance of early lockdown and broad testing in containing the epidemic curve.

## Methods

### Data collection

Data relevant to the number of infected subjects (cases) per day, and number of swabs tested were obtained from daily reports of the "Protezione Civile" and are available at [8]. The

number of inhabitants, population density (persons/km$^2$), and gross domestic product (GDP) *pro capite* (€) were obtained from [9] and are relevant to 2019. The distance between each Region and the origin of the epidemic was determined as the distance (km) between Milano (capital city of Lombardia) and the capital city of each Region.

### Regression model to describe the epidemic dynamics

Generalized logistic curves (Richards model) are used to study the infection trajectories [10]. In this study, we adopted a re-parameterization of the Gompertz growth equation [11] as regression model to describe the cumulative number of infected subjects per day and to define epidemic parameters in Italy and in the Italian Regions. The curve was fitted with the Graph-Pad Prism 8 software by the least square method. The model is: $y = Y_M \cdot (Y_0 / Y_M)^{e^{-kx}}$, where $y$ is the cumulative number of cases, $Y_M$ is the maximum number of cases, $Y_0$ is the number of cases at day 0, $x$ is the time (days) and $k$ is a growth constant. The relative maximum growth rate (i.e. the growth rate at inflection) is given by $k/e$ and the absolute maximum growth rate is given by $Y_M \cdot (k/e)$. The re-parameterization of the Gompertz model adopted, which involves 3 parameters, can be considered as a simpler special case of the 5-parameter Richards model [12, 13], with the inflection (i.e. the point at which the growth rate reaches the maximum to start decreasing subsequently) locked at 36.8% of the upper asymptote (i.e. $Y_M/e$). The 3-parameter Gompertz was preferred to the 5-parameter Richards model for fitting the data according to the Akaike Information Criteria and the Extra-sum-of squares F test.

### Correlation analysis

Using the data obtained from each Region, a correlation analysis was performed to analyze the potential association between "local" or "interventional" variables and the following epidemic parameters: total number of cases, growth rate, inflection time and time to reach 95% and 99% of the total cases. Population density, distance from the origin and GDP were considered as local variables, while number of cases at lockdown (representing the earliness of lockdown implementation with respect to the epidemic trajectory) and number of swabs per case (representing the width of testing) were considered interventional variables. The non-parametric Spearman correlation coefficient "R" was calculated to assess the strength of the correlation. In addition, a multiple regression analysis was performed to verify the independent association between the variables analyzed and the total number of cases predicted. Analyses were performed with GraphPad Prism 8 software.

## Results

### The SARS-CoV-2 epidemic in Italy

The cumulative number of cases per day in Italy and the fitting curve are shown in Fig 1, reporting the absolute number of cases (Fig 1A) and the number of cases/100,000 persons (Fig 1B). The actual number of new cases per day and the number predicted by the curve are shown in Fig 1C (absolute cases) and 1D (cases/100,000 persons). Data were collected until day 81 (May 10) after the first detected case, when the number of cases was 219,070 (363/100,000 persons). The total predicted number of cases is 233,606 (386/100,000 persons). The inflection occurred at day 38 (18 days after lockdown) and the detection of 95% and 99% of the cases (surrogate markers of the potential end of the epidemic) is predicted to occur at day 86 (66 days after lockdown) and at day 113 (93 days after lockdown). The epidemic parameters derived from the model are reported in Table 1. Due to the fact that restrictive measures have been partially loosened on May 4 (day 75), and would be further reduced from May 18 (day 89), the future dynamic of the epidemic might diverge from the curve.

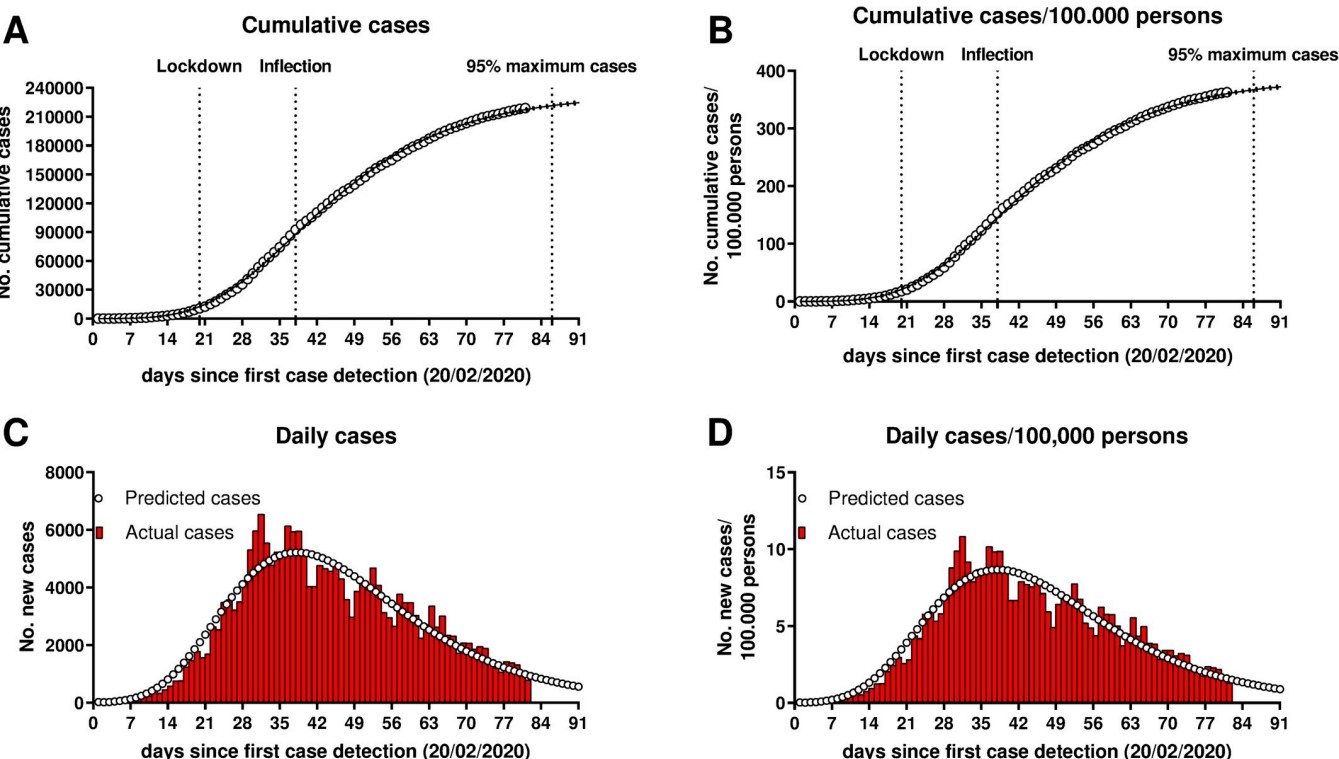

**Fig 1.** Cumulative (A, B) and new daily (C, D) cases of the SARS-CoV-2 epidemic in Italy. In A and B, white circles represent the actual cumulative cases and the black line the regression curve.

The fitting model was also calculated with the data obtained at subsequent time-points during the epidemic: 20, 30, 40, 50, 60 and 70 days (Fig 2). The curves calculated at day 20 (start of lockdown) and 30 (10 days after lockdown) predicted a total number of cases of 3,949 and 1,286/100,000 persons, respectively (about 10 and 4 times higher than the number predicted with the present model), with the inflection point occurring at days 68 and 55, respectively. The curves calculated subsequently predicted a total number of cases between 324 and 407/100,000 persons (with a maximum discrepancy of 16% from the present data indicating 386 total cases) and inflection points between day 35 and day 38. Although sigmoidal growth curves fitted before the inflection point should be considered with caution (and the number of measures available at day 20 and 30 is quite small), these data suggest that the lockdown was effective in flattening the epidemic curve. In addition, the Gompertz curves fitted from day 40 onward (after the inflection point) appears to be a simple and reliable model for the prediction of the dynamic of the epidemic with an acceptable error.

## The SARS-CoV-2 epidemic in the Italian Regions

The dynamic of the epidemic in the Italian Regions is shown in Fig 3 and actual parameters along with those derived from the curves are reported in Table 1. The number of cases at the beginning of the lockdown showed a wide range, from 1 to 58 cases/100,000 persons (median 7), as well as the number of cases at day 81 (58–921, median 258) and the total predicted cases (58–946, median 267). Similarly, the time at which 95% and 99% of the total cases is predicted to occur ranged from day 53 to 104 (median 78) and day 64 to 136 (median 99), respectively. When we observe the epidemic curves (Fig 3A and 3B for a more detailed representation of

**Table 1. Epidemic parameters in Italy and in the Italian Regions.**

| Region | no. cases/100.000 persons (%) at lockdown (day 20) | no. cases /100.000 persons (%) at day 81 | Maximum no. cases /100.000 persons (%) predicted | Maximum growth rate [cases /100.000 persons (%) per day] | Inflection time (day) | Time to reach 95% cases (day) | Time to reach 99% cases (day) |
|---|---|---|---|---|---|---|---|
| V Aosta | 14 (0.01) | 921 (0.92) | 946 (0.95) | 29 (0.03) | 36 | 71 | 91 |
| Trento | 10 (0.01) | 793 (0.79) | 867 (0.87) | 20 (0.02) | 40 | 87 | 112 |
| Lombardia | 58 (0.06) | 810 (0.81) | 847 (0.85) | 18 (0.02) | 35 | 86 | 113 |
| Piemonte | 10 (0.01) | 658 (0.66) | 793 (0.79) | 15 (0.02) | 47 | 104 | 136 |
| Liguria | 9 (0.01) | 567 (0.57) | 639 (0.64) | 13 (0.01) | 43 | 98 | 128 |
| Emilia Romagna | 34 (0.03) | 601 (0.60) | 623 (0.62) | 15 (0.02) | 36 | 80 | 104 |
| Bolzano | 7 (0.01) | 482 (0.48) | 500 (0.50) | 15 (0.02) | 36 | 72 | 92 |
| Marche | 26 (0.03) | 428 (0.42) | 434 (0.43) | 12 (0.01) | 33 | 73 | 95 |
| Veneto | 17 (0.02) | 382 (0.38) | 409 (0.41) | 10 (0.01) | 37 | 83 | 109 |
| Toscana | 7 (0.01) | 262 (0.26) | 276 (0.28) | 7 (0.01) | 38 | 80 | 103 |
| Friuli V G | 10 (0.01) | 258 (0.26) | 267 (0.27) | 7 (0.01) | 36 | 76 | 97 |
| Abruzzo | 3 (0.00) | 237 (0.24) | 251 (0.25) | 6 (0.01) | 40 | 83 | 107 |
| Umbria | 4 (0.00) | 160 (0.16) | 158 (0.16) | 8 (0.01) | 32 | 53 | 64 |
| Lazio | 2 (0.00) | 122 (0.12) | 128 (0.13) | 3 (0.00) | 40 | 85 | 110 |
| Puglia | 1 (0.00) | 107 (0.11) | 113 (0.11) | 3 (0.00) | 40 | 83 | 106 |
| Molise | 5 (0.01) | 121 (0.12) | 106 (0.11) | 3 (0.00) | 38 | 73 | 92 |
| Sardegna | 1 (0.00) | 82 (0.08) | 83 (0.08) | 3 (0.00) | 36 | 69 | 87 |
| Campania | 2 (0.00) | 79 (0.08) | 81 (0.08) | 3 (0.00) | 38 | 73 | 93 |
| Sicilia | 1 (0.00) | 67 (0.07) | 68 (0.07) | 2 (0.00) | 38 | 78 | 99 |
| Basilicata | 1 (0.00) | 68 (0.07) | 67 (0.07) | 3 (0.00) | 36 | 63 | 77 |
| Calabria | 1 (0.00) | 58 (0.06) | 58 (0.06) | 2 (0.00) | 36 | 70 | 89 |
| Median | 7 (0.01) | 258 (0.26) | 267 (0.27) | 7 (0.01) | 37 | 78 | 99 |
| Range | 1–58 (0.00–0.06) | 58–921 (0.06–0.92) | 58–946 (0.06–0.95) | 2–29 (0.00–0.03) | 32–47 | 53–104 | 64–136 |
| Italy | 19 (0.02) | 363 (0.36) | 386 (0.39) | 9 (0.01) | 38 | 86 | 113 |

the initial stage), we can distinguish three clusters that separate after the lockdown: 8 Regions (the most distant from Lombardia, almost all in Southern Italy) showed flatter curves, 4 intermediate and 9 more steepened curves. One Region of the intermediate cluster B (Umbria) showed a faster inflection time and was separated from the other cluster B curves by reaching the plateau earlier and stopping at a number of total cases similar to that of cluster C. It is noteworthy that the Regions with the flatter curves (cluster C) had a median number of 1 (range 1–5) case/100,000 persons at the start of the lockdown, while Regions with the more steepened curves (cluster A) had a median number of 14 (7–58) cases at lockdown. This suggests that starting the lockdown earlier, when the number of cases is still low (i.e. <5 cases/100,000 persons), may have contributed to flattening of the epidemic curve.

## Analysis of the variables associated with the epidemic parameters

The potential association between selected variables and epidemic curve parameters was evaluated by the Spearman's correlation analysis. For each Region, the following "local" variables (see S1 Table) were analyzed: i) distance from the Italian origin of the epidemic (i.e. Lombardia, where the first indigenous cases were detected), ii) population density, iii) GDP *pro capite* (as a rough surrogate indicator of the industrial activity of the Region). In addition, we analyzed the following "interventional" variables: i) number of cases at lockdown (as a marker of the starting point of the lockdown compared to the epidemic curve), and ii) number of swabs

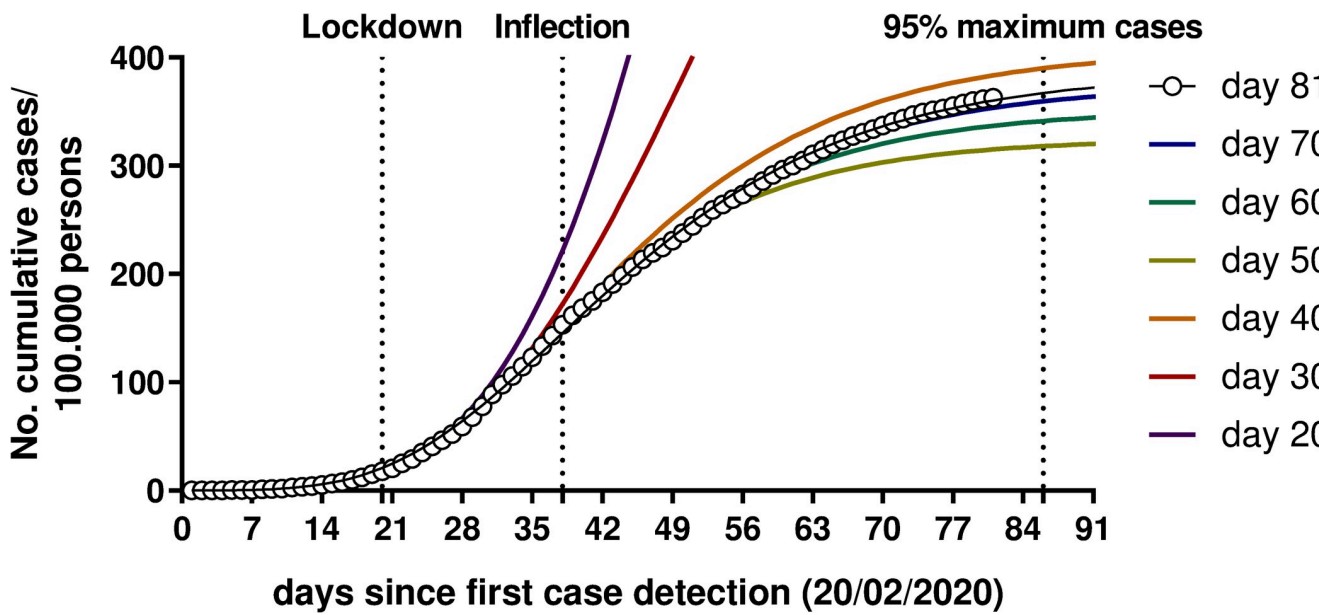

**Fig 2. Determination of the regression curves at different time-points during the epidemic.** White circles represent actual cases.

tested per case detected (as a marker of the extent of testing in the population). The total number of cases and the maximum absolute growth rate were inversely correlated with the distance from the origin and directly with the GDP, but not with the population density. The total number of cases and growth rate were also correlated with the number of cases at lockdown and inversely correlated with the extent of the testing (see Table 2 for details). Times to reach 95% and 99% of the cases were correlated with the distance from the origin, the population density and the extent of the testing (although with less strength). The inflection time did not correlate with any of the variables considered. A correlation plot between the total number of cases and the variables is shown in Fig 4. Similar results were obtained if considering the actual number of cases at day 81 instead of the total number of cases predicted by the curves (S1 Fig). To test whether the selected variables were independently associated with the total number of cases we performed a multiple regression analysis (Table 3 and Fig 5). In particular, the distance from the origin and the GDP were also inversely correlated (Spearman r: -0.81) between each other (i.e. the more distant regions also had a lower GDP, S2 Table). Thus, it is possible that only one of these two variables is independently correlated with the total number of cases. However, the two local variables (distance and GDP) significantly correlated with the total number of cases also in multiple regression analysis (a model with distance, GDP, and their interaction had an $R^2$ of 0.79; Table 3). Similarly, the number of cases at lockdown and the extent of testing were inversely correlated between each other (Spearman r: -0.70, S2 Table). However, the two interventional variables were independently correlated with the total number of cases (model $R^2$: 0.74; Table 3). Lastly, the combination of both local and interventional variables in a model involving distance, GDP and extent of testing showed an $R^2$ of 0.88 in predicting the total number of cases, whereas the number of cases at lockdown did not significantly improve the model.

## Discussion

In this study, we used the Gompertz model to describe the SARS-CoV-2 epidemic in Italy and in the Italian Regions. The curve parameters obtained were adopted to analyze the differences

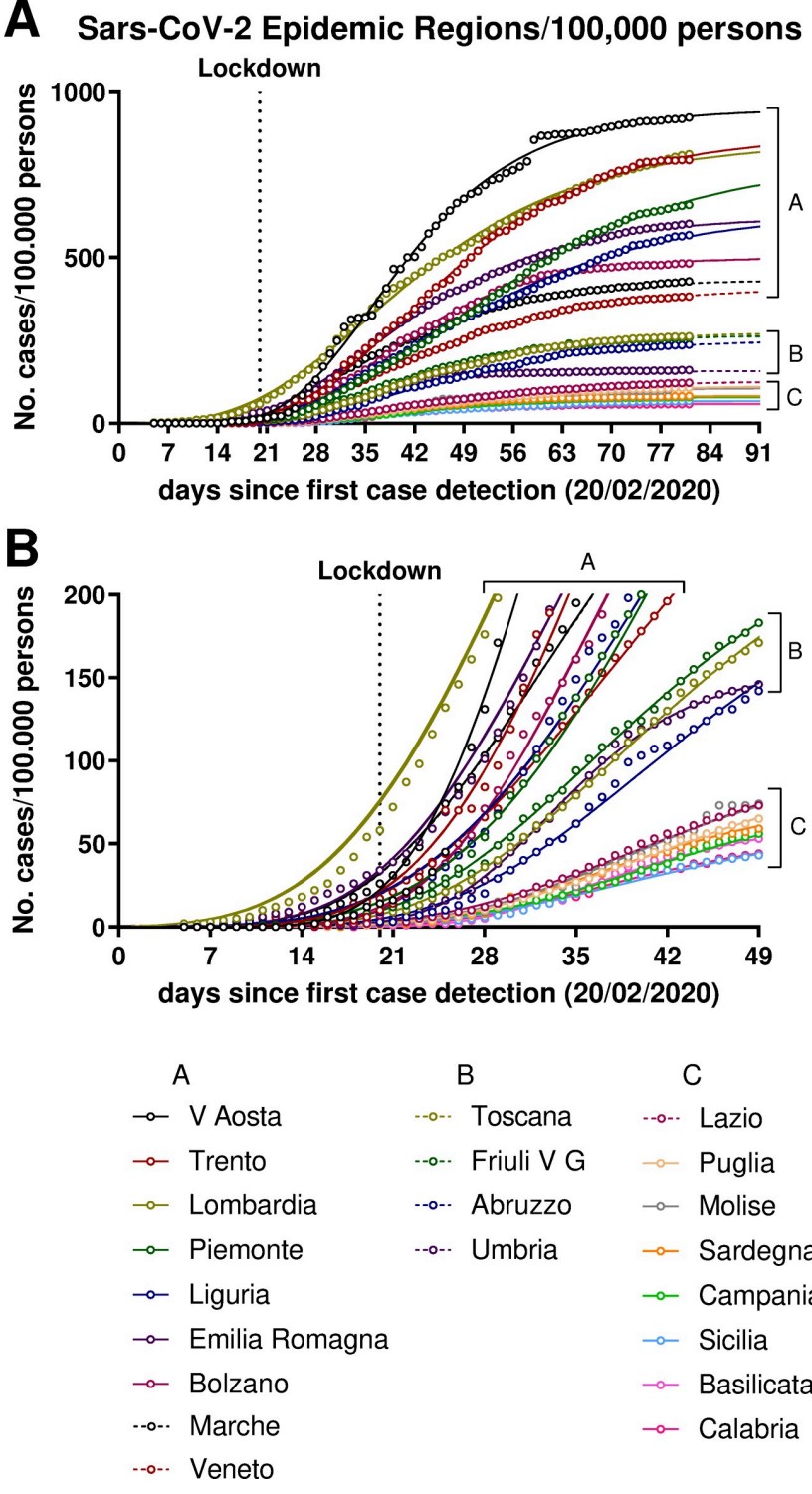

**Fig 3.** Cumulative cases of the SARS-CoV-2 epidemic in the Italian Regions (A). In panel (B) a magnification of the initial phase is shown.

in the infection spreading within the Italian Regions and the relevant associated variables. The model adopted was simple and provided consistent results when calculated at different time-

**Table 2. Correlation coefficient R (p value) between epidemic parameters and local or interventional variables.**

| Parameter | Distance from the origin | Population density | Gross Domestic Product *pro capite* | No. cases at lockdown | No. swabs per case |
|---|---|---|---|---|---|
| Total no. cases/ 100.000 persons | -0.94 (<0.001) | ns | 0.78 (<0.001) | 0.86 (<0.001) | -0.86 (<0.001) |
| Maximum growth rate (cases/100.000 persons per day) | -0.92 (<0.001) | ns | 0.80 (<0.001) | 0.87 (<0.001) | -0.76 (<0.001) |
| Inflection time | ns | ns | ns | ns | ns |
| Time to reach 95% of the cases | -0.55 (0.010) | 0.63 (0.002) | ns | ns | -0.66 (0.001) |
| Time to reach 99% of the cases | -0.57 (0.007) | 0.65 (0.002) | ns | 0.43 (0.051) | -0.70 (<0.001) |

points after the inflection time. Limitations of this study concern the likely underestimation of the actual number of cases by the reported data (testing was indicated for symptomatic subjects only), as well as potential differences in the lapse of time between testing and data

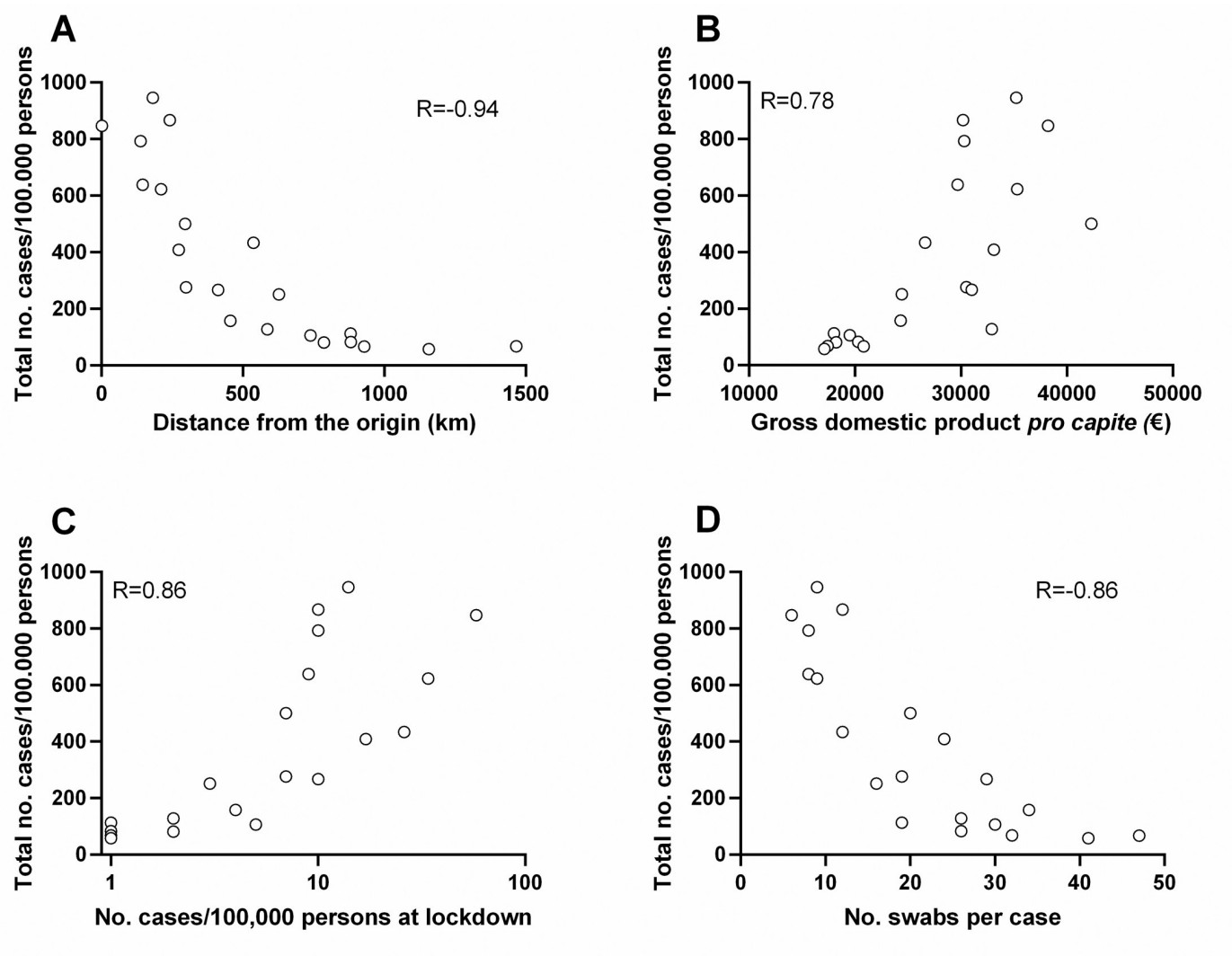

**Fig 4. Correlation between the total number of cases and local or interventional variables.** Correlation between the total number of cases predicted and (A) the distance from the origin of the epidemic, (B) gross domestic product *pro capite*, (C) number of cases at lockdown (note that the x-axis is logarithmic) and (D) number of swabs tested per case detected.

**Table 3. Multiple linear regression analysis of local, interventional variables, and their combination *vs* total number of cases.**

| Model | Variable | Estimate parameter value | p | R² |
|---|---|---|---|---|
| Local | GDP | 0.026 | <0.001 | |
| | Distance | 0.673 | 0.028 | |
| | Interaction (GDP x Distance) | $-5.78 \times 10^{-5}$ | 0.002 | 0.79 |
| Interventional | Intercept | 489.2 | 0. 010 | |
| | Swabs | -13.63 | 0.011 | |
| | Cases at Lockdown | 234.8 | 0.035 | 0.74 |
| Local + Interventional | GDP | 0.030 | <0.001 | |
| | Interaction (GDP x Distance) | $-2.04 \times 10^{-5}$ | 0.013 | |
| | Interaction (GDP x Swabs) | $-6.23 \times 10^{-4}$ | <0.001 | |
| | Interaction (Distance x Swabs) | 0.008 | 0.036 | 0.88 |

GDP, Gross Domestic Product.

reporting among the Regions. The rate of infected subjects in Italy was estimated to be at least ten times higher than that diagnosed [14]. However, with the assumption that the fraction of undiagnosed cases is constant, the general dynamic of the event here described is reliable (although the absolute epidemic extent is not available), and regression analysis helps in reducing stochastic measurement errors.

We analyzed the potential association between local and interventional factors with the uneven spreading of the epidemic in the Italian Regions. The local factors analyzed are the population density of the region, the GDP (as a surrogate indicator of industrial activity) and the distance from the origin of the epidemic. The interventional factors analyzed were lockdown implementation and the extent of testing.

The population density was not associated with the total number of cases predicted and with the growth rate. This observation may reflect the fact that Regions with a low average population density may have large territories with low population density and highly populated urban centers, where the infection can spread easily. On the other hand, the correlation between the extent of the epidemic and the GDP may indicate that, before lockdown implementation, the infection spread more efficiently in the areas with higher industrialization, probably because of broader social interactions and population movement associated with the

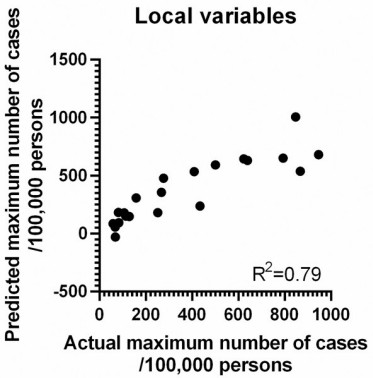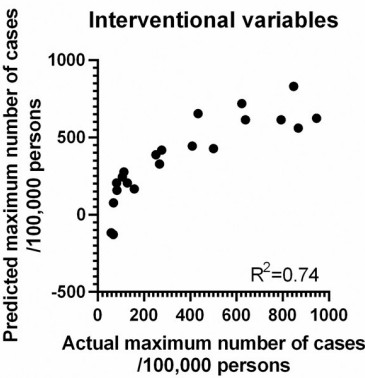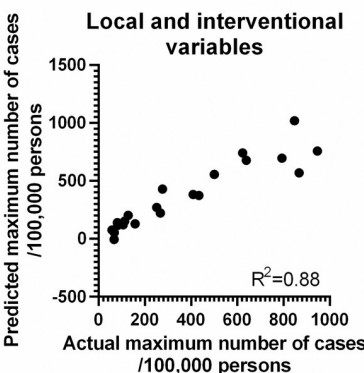

**Fig 5. Multiple linear regression analysis of local, interventional variables, and their combination *vs* the total number of cases.** The actual number of cases is reported on the x-axis and the number predicted by the regression lines are reported on the y-axis. Variables used in the three regression analyses are reported in Table 3.

industrial activities. Lastly, the epidemic was limited in the Regions that were more distant from the initial foci. This could be also a consequence of the efficacy of the containment measures. In fact, we can hypothesize that the epidemic would have spread similarly in all the areas, although with delayed kinetics, if containment measures would have not been implemented. In this respect, the WHO advised to slow and stop transmission, prevent outbreaks and reduce case numbers and, in response to large outbreaks of community transmission, promoted the practice of social distancing among the public health measures [3].

In our analysis, as indicators of the interventional factors, we considered the number of cases at the start of the lockdown (as marker of the earliness of the measure's implementation) and the number of swabs tested per case detected (as a marker of the extent of testing). The number of infected cases at the start of the lockdown was associated with the local factors discussed above (i.e. Regions with higher GDP and lower distance from the origin had a higher number of cases at lockdown). Where the number of infected subjects was lower at the start of lockdown, the total number of infected subjects predicted for the end of the epidemic also remained lower. Thus, the restrictive measures adopted in Italy appear effective in reducing the transmission of the infection and flattened the epidemic curve, as shown also by mathematical modelling [15, 16], and their early application is associated with a better control of the spread of the infection. Another indirect indication of the efficacy of the measures adopted is the observation that the curves determined using only the data obtained before or a few days after the lockdown predicted a higher number of infected subjects than the curves determined from 20 days after the lockdown onwards. Although this observation may be questionable, due to both the low number of data used to fit the model at the beginning of the epidemic, as well as the difficulty in defining the curve parameters before the inflection point, there was a clear trend towards a flattening of the curves after the lockdown.

In addition, a broader testing (considered here as a higher number of swabs tested per case detected) was also associated with a better control of the epidemic. Since this is an observational retrospective analysis, we cannot prove a causal relationship between wider testing and the containment of the epidemic. It is also possible that in the areas with higher frequency of infected subjects, the testing capacity of the health system was overwhelmed. In this respect, the WHO guidance suggested to prioritize testing in health care settings and vulnerable groups and to test only the first suspected cases in closed settings [3]. Nevertheless, the correlation between wider testing and lower frequency of cases is in agreement with results from a model study by Giordano and colleagues [16], which showed, through simulation analysis, that widespread testing helps in reducing the epidemic burden. Another model showed that timely diagnosis of infection shortens the peak time, decreases the peak value of new infections, and reduces the number of cumulative infections [17]. A wide testing policy permits a rapid identification and isolation of infected subjects, thus helping to limit the epidemic.

Broader testing may also have the following advantages. Widespread testing could be combined with softer social-distancing measures, thus obtaining an effect similar to a stricter lockdown in limiting the epidemic [16]. In addition, reduction in the total number of infected subjects will also reduce the number of those patients that require admission to intensive care units.

In view of the rise of secondary epidemic waves in Italy and in other countries, healthcare policymakers could consider increasing testing capacity and implementing broad screening as a potentially cost-effective containment option that would permit a milder lockdown and a reduced need for intensive care units (thus reducing the relevant costs). Broad screening would allow rapid detection of new epidemic foci, identification and isolation of infected subjects, prompt closure of the affected area and, finally, an earlier resolution of epidemic episodes

with a lower number of infected subjects. It is worth considering that undocumented infection was found to be the cause of the large spread of the infection [18].

With respect to testing possibilities, rapid molecular methods providing reliable results have been developed [19]. Rapid serological tests have also been proposed [20] and may be useful as first-line screening in the territory, although they are not sufficiently accurate for diagnosis of hospitalized patients with acute infection [21]. As for neutralizing assays, it would be important to verify where non-antibody serum inhibitor factors present in human sera and neutralizing human CoV OC43 may interfere with neutralizing assays for SARS-CoV-2 [22].

In conclusion, our analysis of the first SARS-CoV-2 epidemic evolution in Italy could help in the planning of future strategies for the control of secondary breakthrough episodes. In fact, in agreement with previous simulation studies, the analysis of the actual epidemic data support the effectiveness of interventional measures such as population-wide testing and early lockdown enforcement in limiting the burden of the SARS-CoV-2 epidemic. Lockdown appears more effective if applied at the very beginning of the epidemic, when the number of cases is low. On the other hand, the implementation of a broad testing capacity also appears effective and may have the advantage of reducing the need for social distancing and closure of the productive activities required by lockdown enforcement.

## Supporting information

**S1 Fig. Correlation between the number of cases at day 81 and local or interventional variables.** Correlation between the total number of cases predicted and (A) distance from the origin of the epidemic, (B) gross domestic product *pro capite*, (C) number of cases at lockdown (note that the x-axis is logarithmic) and (D) number of swabs tested per case detected. (TIF)

**S1 Table. Local variables analyzed for the association with epidemic parameters.** (DOCX)

**S2 Table. Correlation matrix between local and interventional variables.** (DOCX)

## Acknowledgments

We thank Antonio Colangelo for helpful discussion.

## Author Contributions

**Conceptualization:** Daniele Lilleri.

**Data curation:** Federica Zavaglio, Elisa Gabanti.

**Formal analysis:** Daniele Lilleri.

**Methodology:** Daniele Lilleri.

**Writing – original draft:** Daniele Lilleri.

**Writing – review & editing:** Federica Zavaglio, Giuseppe Gerna, Eloisa Arbustini.

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
