## [Decision Letter · Decision Letter 0]

2 Oct 2020

PONE-D-20-19843

Analysis of SARS-CoV-2 epidemic in Italy: role of baseline and interventional factors in the epidemic control.

PLOS ONE

Dear Dr. Lilleri,

Thank you for submitting your manuscript to PLOS ONE. After careful consideration, we feel that it has merit but does not fully meet PLOS ONE’s publication criteria as it currently stands. Therefore, we invite you to submit a revised version of the manuscript that addresses the points raised during the review process.

We apologize for the delay as it has been extremely difficult in obtaining reviewers, as most people working on COVID-19 have been too busy. The manuscript in its current format requires revision. As indicated by the reviewers, there needs to be more description and justification of the methods and analytical models used. 

We look forward to receiving your revised manuscript.

Kind regards,

Patrick Tang, M.D., Ph.D.

Academic Editor

PLOS ONE

Journal Requirements:

'This work was partially funded by Fondazione Cariplo, Milano, Italy (Grant CoVIM to DL). The funder had no role in study design, data collection and analysis, decision to publish, or preparation of the manuscript.'

a. Please provide an amended statement that declares *all* the funding or sources of support (whether external or internal to your organization) received during this study, as detailed online in our guide for authors at http://journals.plos.org/plosone/s/submit-now

Please also include the statement “There was no additional external funding received for this study.” in your updated Funding Statement.

Reviewers' comments:

Reviewer's Responses to Questions

**Comments to the Author**

1. Is the manuscript technically sound, and do the data support the conclusions?

Reviewer #1: Partly

Reviewer #2: Yes

2. Has the statistical analysis been performed appropriately and rigorously? 

Reviewer #1: Yes

Reviewer #2: Yes

3. Have the authors made all data underlying the findings in their manuscript fully available?

Reviewer #1: Yes

Reviewer #2: Yes

4. Is the manuscript presented in an intelligible fashion and written in standard English?

Reviewer #1: No

Reviewer #2: Yes

5. Review Comments to the Author

Reviewer #1: This manuscript is about the Analysis of SARS-CoV-2 epidemic in Italy: role of baseline and intervention factors in

the epidemic control.

In as much as SARS-CoV-2 manuscripts are of interest, I find almost every aspect of the paper needs further revision for clarity; context; methodological rigor; and interpretation. Please see my comments below:

Firstly. Please find native speaker to assist with the English language editing and flow of the manuscript.

Abstract

Please give a brief introduction of what your paper seeks to achieve (aims and objectives of your paper)

Also, please talk about the methodology used. for example how your sample was achieved and how you interpreted your results. for example, talk about cases were represented in xxx per 10,000 and xxxx was shown in means, or percentages to make your work more readable to the international world.

At the bottom part of your abstract, please talk a little on conclusion.

Introduction

Your introduction is too scanty. though CoV-2 is a novel disease, I believe you can deliberate more on literature in both Europe and around the globe than just talking about how lock downs and restrictions were done in Italy which is already known because of news casters.

please talk about the disease, its global burden ( for example, in USA, China, Africa, etc and limit it to Italy and compare these burdens with that of Italy. compare the international strategies of other countries with what happened in Italy. Please explore more on this.

Methods

In your methods thank you for telling us about how 'Swab' reports were taken and the availability of the data. However, why did you decide to use the Gompertz growth equation ( which is a nonlinear regression model, used to describe growth curves

mostly in biology to determine growth of animals and plants, as well as the number or volume of bacteria and cancer cells?. Can you please explain further on why this is befitting for your work, or replace it with linear curves if possible.Also, please explain more on the correlations between the variables.

Table 1: please add percentages

You seem to be dwelling more on the GDP of the various regions, please represent that with a graph where appropriate and compare that with the various interventions within the regions in the discussion part.

Discussions

please discuss you findings as in: What was the baseline, what were the intervention factors, are they attributed to GDPs of the various regions? What are the implications, and how will that be used as a future baseline implementation, should there be any future epidemic. lastly, please also discuss a bit on how the CoV-2 protocol approved by the WHO either helped or did not in the intervention factors to the control of the epidemic in Italy.

Thank You.

Reviewer #2: Thanks for developing this nice manuscript. Authors analyzed the impact of baseline and interventional variables on the epidemic curve in each region in Italy. They found that the number of cases correlated inversely with the distance from the area in which first cases were detected and directly also with the gross domestic product pro capite of the Region. In their opinion, earlier start of the lockdown and a wider testing were associated with a lower final number of total cases, which could help policy makers to act promptly.

My only comment on the analysis technique: did you consider the probability of reinfection in your model? It may be true that 99% of the population will got Covid by 113 days in Italy, but if reinfection occurs after around one month for some of the recovered patients, how we will fit that in our model? Please respond or adjust your analysis.

6. PLOS authors have the option to publish the peer review history of their article (what does this mean?). If published, this will include your full peer review and any attached files.

Reviewer #1: No

Reviewer #2: **Yes: **Mahbub-Ul Alam

---

## [Author Response · Author response to Decision Letter 0]

26 Oct 2020

We believe to have addressed the issues raised by the reviewers, which helped in improving the manuscript. We added a supporting S1 Figure reporting the data that were referred as “data not shown” in the previous version. In addition, we would like to update the funding statement as follows:

“This work was funded by Fondazione Cariplo, Milano, Italy (Grant CoVIM to DL). The funder had no role in study design, data collection and analysis, decision to publish, or preparation of the manuscript. There was no additional external funding received for this study.”

Please find below or response to reviewers' comments.

Reviewer #1: This manuscript is about the Analysis of SARS-CoV-2 epidemic in Italy: role of baseline and intervention factors in the epidemic control.

In as much as SARS-CoV-2 manuscripts are of interest, I find almost every aspect of the paper needs further revision for clarity; context; methodological rigor; and interpretation. Please see my comments below:

Firstly. Please find native speaker to assist with the English language editing and flow of the manuscript.

R: The manuscript was revised by a native speaker

Abstract

Please give a brief introduction of what your paper seeks to achieve (aims and objectives of your paper)

Also, please talk about the methodology used. for example how your sample was achieved and how you interpreted your results. for example, talk about cases were represented in xxx per 10,000 and xxxx was shown in means, or percentages to make your work more readable to the international world.

At the bottom part of your abstract, please talk a little on conclusion.

R: we thank the Reviewer for his suggestion and revised the abstract accordingly.

Introduction

Your introduction is too scanty. though CoV-2 is a novel disease, I believe you can deliberate more on literature in both Europe and around the globe than just talking about how lock downs and restrictions were done in Italy which is already known because of news casters.

please talk about the disease, its global burden ( for example, in USA, China, Africa, etc and limit it to Italy and compare these burdens with that of Italy. compare the international strategies of other countries with what happened in Italy. Please explore more on this.

R: On the basis of the reviewer suggestion, the Introduction was improved by adding information about the disease burden throughout the world, the major clinical characteristics of the disease, and a comparison of the international containment strategies adopted, quantified according to the Oxford stringency score.

Methods

In your methods thank you for telling us about how 'Swab' reports were taken and the availability of the data. However, why did you decide to use the Gompertz growth equation ( which is a nonlinear regression model, used to describe growth curves

mostly in biology to determine growth of animals and plants, as well as the number or volume of bacteria and cancer cells?. Can you please explain further on why this is befitting for your work, or replace it with linear curves if possible. Also, please explain more on the correlations between the variables.

R: The epidemic trajectory follow a sigmoidal curve (logistic model), thus a linear regression model is not suitable for the analysis. Usually the 5-parameters Richards curve is adopted. We used a re-parametrization of the Gompertz growth equation which is a special case of the Richards equation. The Gompertz curve has a lower number of parameters than the Richards curve (3 instead of 5): thus, for the purpose of this study, the simpler model was preferred (according to both to Akaike Information Criteria and Extra-sum-of squares F test) because is easier to fit while providing an equally acceptable description of the epidemic trajectory. In addition, the parameters analysed in this study (maximum number of infected subjects and maximum growth rate) are easily obtained from the fitted curve. We explained more on it as well as on the correlation analysis in the relevant Methods paragraphs.

Table 1: please add percentages

R: percentages have been added to Table 1

You seem to be dwelling more on the GDP of the various regions, please represent that with a graph where appropriate and compare that with the various interventions within the regions in the discussion part.

R: we added a supplementary S1 Table, showing population density, distance and GDP for each Region, and S2 Table showing the correlation matrix between the variables analyzed in multiple regression (GDP, distance, no. cases at lockdown, no. swabs per case). In our study, GDP was considered as a surrogate marker of the industrial activity of the regions, which appears to have influenced the extent of the epidemic before lockdown implementation. The infection may have spread more efficiently in the areas with higher industrialization, probably because of broader social interactions and population movement associated with the industrial activities.

Discussions

please discuss you findings as in: What was the baseline, what were the intervention factors, are they attributed to GDPs of the various regions? What are the implications, and how will that be used as a future baseline implementation, should there be any future epidemic. lastly, please also discuss a bit on how the CoV-2 protocol approved by the WHO either helped or did not in the intervention factors to the control of the epidemic in Italy.

Thank You.

We thank the reviewer for the suggestions on Discussion improvement. We replaced the term “baseline” factor with “local” factors, which appears more appropriate. We defined what are the local and interventional factors considered (para 2), then we discussed about the role of local factors (para 3) and about the role of interventional factors and their association with GDP and distance (paras 4-6). In the last paras (7-9) we discussed about implications of our analysis for the management of secondary waves or future epidemic. We mentioned also the WHO interim guidance on response actions for COVID-19 both in the Introduction and in the Discussion. We believe that now the Discussion is clearer.

Reviewer #2: Thanks for developing this nice manuscript. Authors analyzed the impact of baseline and interventional variables on the epidemic curve in each region in Italy. They found that the number of cases correlated inversely with the distance from the area in which first cases were detected and directly also with the gross domestic product pro capite of the Region. In their opinion, earlier start of the lockdown and a wider testing were associated with a lower final number of total cases, which could help policy makers to act promptly.

My only comment on the analysis technique: did you consider the probability of reinfection in your model? It may be true that 99% of the population will got Covid by 113 days in Italy, but if reinfection occurs after around one month for some of the recovered patients, how we will fit that in our model? Please respond or adjust your analysis.

R: we appreciate the reviewer’s considerations about our manuscript. The model does not consider the probability of reinfection but only new cases of infection. However, there was no case of reinfection reported in Italy during the period analyzed.

---

## [Editor Report · Decision Letter 1]

2 Nov 2020

Analysis of the SARS-CoV-2 epidemic in Italy: the role of local and interventional factors in the control of the epidemic.

PONE-D-20-19843R1

Dear Dr. Lilleri,

We’re pleased to inform you that your manuscript has been judged scientifically suitable for publication and will be formally accepted for publication once it meets all outstanding technical requirements.

Kind regards,

Patrick Tang, M.D., Ph.D.

Academic Editor

PLOS ONE
---

## [Editor Report · Acceptance letter]

4 Nov 2020

PONE-D-20-19843R1 

Analysis of the SARS-CoV-2 epidemic in Italy: the role of local and interventional factors in the control of the epidemic. 

Dear Dr. Lilleri:

I'm pleased to inform you that your manuscript has been deemed suitable for publication in PLOS ONE. Congratulations! Your manuscript is now with our production department. 

Kind regards, 

on behalf of

Dr. Patrick Tang 

Academic Editor

PLOS ONE